# Survey of Deep-Learning Approaches for Remote Sensing Observation Enhancement

**DOI:** 10.3390/s19183929

**Published:** 2019-09-12

**Authors:** Grigorios Tsagkatakis, Anastasia Aidini, Konstantina Fotiadou, Michalis Giannopoulos, Anastasia Pentari, Panagiotis Tsakalides

**Affiliations:** 1Signal Processing Lab (SPL), Institute of Computer Science, Foundation for Research and Technology-Hellas (FORTH), 70013 Crete, Greece; aidini@csd.uoc.gr (A.A.); kfot@ics.forth.gr (K.F.); mgiannop@ics.forth.gr (M.G.); anpentari@gmail.com (A.P.); tsakalid@ics.forth.gr (P.T.); 2Computer Science Department, University of Crete, 70013 Crete, Greece

**Keywords:** deep learning, convolutional neural networks, generative adversarial networks, super-resolution, denoising, pan-sharpening, fusion, earth observations, satellite imaging

## Abstract

Deep Learning, and Deep Neural Networks in particular, have established themselves as the new norm in signal and data processing, achieving state-of-the-art performance in image, audio, and natural language understanding. In remote sensing, a large body of research has been devoted to the application of deep learning for typical supervised learning tasks such as classification. Less yet equally important effort has also been allocated to addressing the challenges associated with the enhancement of low-quality observations from remote sensing platforms. Addressing such channels is of paramount importance, both in itself, since high-altitude imaging, environmental conditions, and imaging systems trade-offs lead to low-quality observation, as well as to facilitate subsequent analysis, such as classification and detection. In this paper, we provide a comprehensive review of deep-learning methods for the enhancement of remote sensing observations, focusing on critical tasks including single and multi-band super-resolution, denoising, restoration, pan-sharpening, and fusion, among others. In addition to the detailed analysis and comparison of recently presented approaches, different research avenues which could be explored in the future are also discussed.

## 1. Introduction

Remote sensing of the environment, and especially Earth observation, is witnessing an explosion in terms of volume of available observations, which offer unprecedented capabilities towards the global-scale monitoring of natural and artificial processes [1,2]. However, the increase in volume, variety and complexity of measurements has led to a situation where data analysis is causing a bottleneck in the observation-to-knowledge pipeline. To address this challenge, machine learning approaches have been introduced for automating the analysis and facilitating the enhancement of remote sensing observations [3]. Unlike conventional machine learning approaches which first perform the extraction of appropriate hand-crafted features and then apply shallow classification techniques, deep machine learning, and Deep Neural Networks (DNNs) in particular, have demonstrated astounding capabilities, which is primarily attributed to the automated extraction of meaningful features, removing the need for identifying case-specific features [4]. The driving force behind the success of DNNs in image analysis can be traced to the following three key factors.

More data available for training DNNs, especially for cases of supervised learning such as classification, where annotations are typically provided by users.More processing power, and especially the explosion in availability of Graphical Processing Units (GPUs) which are optimized for high through-processing of parallelizable problems such as training DNNs.More advanced algorithms which have allowed DNNs to grow considerably both in terms of depth and output dimensions, leading to superior performance compared to more traditional shallow architectures.

The advent of DNNs has led to a paradigm shift in the remote sensing data analysis [5], where significant effort has been given in applying DNN method for supervised learning problems in imaging, such as multi-class [6] and multi-label [7] image classification, autonomous vehicle operations [8], and accelerating magnetic resonance imaging [9] among others. While the problem of supervised classification of remote sensing observations has been under intense investigation in the past four years, in this work we consider the less explored situations involving remote sensing observation enhancement, which can be broadly considered to be instances of inverse imaging problems.

In inverse imaging problems, the objective is to recover high-quality signals from degraded, lower quality observations by inverting the effects of the degradation operation [10]. Indicative cases of inverse problems in optical remote sensing include estimation of higher spatial, spectral and temporal resolution observations, restoration through removal of noise, and enhancement of observation quality and characteristics through fusion of observations from different modalities. The challenge in inverse imaging problems lays in the fact that these problems are by nature ill-posed, i.e., a large number of different high-resolution images (potentially infinite) can be mapped to the same low-quality observations. As such, inverting the process amounts to properly exploiting any prior knowledge which might be available such as sparsity of representation or statistical priors [11].

In recent years, a large number of data-driven approaches have been presented for handling inverse problems where prior knowledge is automatically extracted through training machine learning systems with input-output pairs, while in the past few years, the machine learning approach of choice for this class of problems has been the deep-learning framework, and prolific architectures such as Convolutional Neural Networks (CNNs) [12]. Unlike typical natural images, in the context of remote sensing, several specific challenges are present and must be addressed including:High dimensionality of observations where Multispectral (MS) and Hyperspectral (HS) are often the input and thus exploiting spatial and spectral correlations is of paramount importance.Massive amounts of information encoded in each observation due to the large distance between sensor and scene, e.g., 400–600 km for low Earth orbit satellites, which implies that significantly more content is encoded in each image compared to a typical natural image.Sensor specific characteristics, including the radiometric resolution which unlike typical 8-bit imagery, in many cases involves observations of 12-bits per pixel.Challenging imaging conditions which are adverse affected by environmental conditions including the impact of the atmospheric effects such as clouds on the acquired imagery.

In this paper, we focus on passive imaging remote sensing platform, the majority of which involves multispectral and hyperspectral imaging instrument among spaceborne satellites and explore how DNNs can address associated inverse imaging problems targeting the quality enhancement of the acquired imagery. The rest of this paper is organized as follows: In Section 2, the prototypical DNN architectures considered in the related literature are presented including Convolutional Neural Networks (CNN), Autoencoders (AE), and Generative Adversarial Networks (GAN). In Section 3 we focus on the problem of super-resolution observation from either color, multispectral or hyperspectral observations while in Section 4 we explore methods related to the problem of pan-sharpening. Section 5 present methodologies for denoising observations and estimating missing measurements while Section 6 outlines the state-of-the-art in fusion of diverse sources, including the case of fusion of observation from different modalities. In Section 7, a set of challenges related to the problems at hand are presented, while future research endeavors and research endeavors which cannot be directly mapped to existing classes are presented. Please note that in this work, we focus exclusively on the enhancement of remote sensing observations, acquired either by an airborne, e.g., unmanned aerial vehicles, or spaceborne satellites, and the observations are single images or image sequences (video) with one, three or multiple spectral bands.

## 2. Deep Neural Network Paradigms

In this section, we provide an overview of the three major DNN approaches related to remote sensing image enhancement, namely Autoencoders (AE), Convolutional Neural Networks (CNN), and Generative Adversarial Networks (GANs).

### 2.1. Convolutional Neural Networks (CNNs)

CNNs are specific DNN architectures which involve the use of convolutions, instead of the more traditional matrix multiplications, in several layers, and are an ideal tool for processing regularly sampled data such as 2D and 3D imagery. Prototypical CNN architectures for image analysis tasks are composed of four key layers types, namely convolutional layers, nonlinear activations, pooling and fully connected layers [13].

#### 2.1.1. Key Components of CNNs

*Convolutional layers*. For the case of 2D imagery, the convolution operation applied on input features I(i,j,k), where (i,j) indicate the spatial location and *k* the input channel, is expressed as
$$Y^{\left(l\right)}=I^{\left(k\right)}*K^{\left(l,k\right)}=\sum_{m}\sum_{n}\sum_{k}I\left(m,n;k\right)K\left(i-m,j-n;l,k\right)$$
where $K^{\left(l,k\right)}\in R^{m\times n}$ is the convolution kernel of size m×n, associated with input channel *k* and output channel *l*. In typical setups, multiple kernels are learned for each convolutional layer. The key motivation of using convolutional layers is their ability to provide translation invariance with respect to the detected features. Furthermore, the much smaller size of the kernels compared to the input also promotes parameter sharing and sparsity in terms of connectivity, which also has a positive impact in terms of storage requirements. Two additional parameters in the design of convolutional layers, moreover the size of the kernel, are the step size of its application, a parameter known as stride and the method of handling the boundaries.

*Activation function*. Each feature generated by convolving the inputs from the previous layer with a kernel is then passed through a nonlinear function called activation function. Although more traditional CNN realizations employed activations such as the sigmoid and hyperbolic tangent (tanh), the majority of modern approaches use the Rectified linear unit, ReLU, which essentially imposes a non-negatively constraint, i.e., $\hat{x}=max\left(0,x\right)$ and variations such as the parametric PReLU [14] where ${\hat{x}}_{i}=max\left(0,x_{i}\right)+a_{i}\min\left(o,x_{i}\right)$ for the *i*-th channel. In addition to superior performance, ReLU variants can also address the issues of vanishing gradient in deep architectures.

*Transposed convolution (or deconvolution) layers*. A requirement of CNN when applied in problems such as super-resolution is the increase in spatial resolution of the original input image, which has been tackled through two approaches, (i) by performing an initial interpolation so that the DNN is responsible for estimating the missing high frequencies, and (ii) by directly operating on the low spatial resolution image and introducing deconvolution layers before producing the enhanced image [15]. While in convolutional layers, the stride, i.e., the step size for the application of the kernel, is a integer number greater or equal to one, in deconvolutional layers a fractional stride value is used, which in practice can still be realized through typical convolution operations using appropriate padding or by reshuffling the outputs of the previous layer [16].

A key characteristic associated with CNN is the receptive field, a termed borrowed from the biology which refers to the part of the sensed space that can elicit some neural response. Receptive fields apply primarily to the case of CNN and refer to the spatial extend associated with each convolution kernel, while in the majority of CNN architectures for image enhancement, both generic and remote sensing ones, the convolution kernels are of size 3 × 3, and in some cases 5 × 5 or even 7 × 7. However, due to the hierarchical structure of CNNs, each subsequent layer has a larger effective receptive field compared to the previous layer, allowing the CNN to progressively consider larger and larger parts of the input image. This process increases the receptive field by a factor of two for each additional layer. Dilated convolutions can also be used for increasing the receptive field by skipping pixels during their applications, e.g., a 3 × 3 kernel is effectively applied on a 5 × 5 region by padding, allowing for faster rates of increase. That is not to say that pooling operators also increase the receptive field; however, they are not typically used in inverse problem due to the unwanted reduction is the output size. In terms of image enhancement, it has been shown that increasing the size of the receptive field, e.g., through the use of dilated convolutions, can lead to the extraction of more important features for generic image denoising [17] and super-resolution [18], as well as in despeckling synthetic aperture radar (SAR) observations [19].

For completeness we note that in addition to the aforementioned layers, two other types of layers are common in the CNN literature, namely pooling and fully connected layers [13]. To reduce the size of the feature space and provide additional invariances, pooling layers, such as max pooling and average pooling, are introduced for aggregating the responses from small spatial regions and propagating the maximum or the average response respectively to subsequent layers. In fully connected layers, each node is connected with every other node from the previous layer, following the prescription of typical Multi-layer Perception architectures [20]. Fully connected and pooling layers are more frequently found in scenarios involving classification tasks where sequences of such layers are cascaded reaching the final output layer, e.g., [21,22,23]; however, since reduction of dimensionality is not required in inverse imaging problems, they are not typically employed for observation enhancement tasks.

#### 2.1.2. Training and Optimization of CNNs

The process of training a CNN, i.e., identifying the optimal values for the kernel weights in the convolution and potentially fully connected or other layers, involves two major design choices, the selection of the appropriate loss function and the selection of the optimization process. Formally, given the CNN architecture parameters $h_{\theta }$ and the input and target variables x,y, the loss function $L\;=\;E(h_{\theta }\left(x\right)-y)$ is typically the $\ell_{2}$ norm between the predicted and the target high-quality observation. The fundamental algorithm for training DNN is the Stochastic Gradient Descend (SGD) [24]; however, state-of-the-art variations of SGD such as the Adam [25] and the AdaDelta [26] offer very similar performance at a much lower computational cost.

While increasing the number of training examples typically leads to better performance, a challenging caveat is that the network can be too adapted to the training example and fail when new, unseen examples, are presented. This phenomenon is called overfitting and special measures must be introduced to combat it. To address this phenomenon, two important mechanisms have been proposed, namely dropout and regularization. In many cases, obtaining a enough training examples is not possible, or it is very time and resource consuming. To address this situation, one approach involves training a DNN using a similar yet much larger dataset and then fine-tuning the network with a much more limited number of case-specific examples. This process is known as transfer learning and its use in deep-learning model [27] and has been instrumental for classification problems where a limited number of training examples is available.

Dropout [28] is a very successful method for addressing the problem of overfitting in terms of performance, yet extremely light in terms of computational requirements. This technique amounts to deactivating several nodes during each iteration of the training process. This forces the network of learn alternative paths for predicting the correct output, thus leading to greater flexibility in terms of generalization capacity. Although dropout can be introduced at any stage of training, the standard practice is to introduce it in the fully connected layers since these layers account for the bulk of the trainable parameters.

Batch Normalization is a local (per layer) normalizer that operates on the node activations in a way similar to the initial normalizing technique applied to the input data in the pre-processing step, and studies [29] have shown that it can work very effectively as a method for addressing overfitting. The primary goal of Batch Normalization is to enforce a zero mean and standard deviation of one for all activations of the given layer and for each mini-batch. The main intuition behind Batch Normalization lies in the fact that as the neural network deepens, it becomes more probable that the neuronal activations of intermediate layers might diverge significantly from desirable values and tend towards saturation. This is known as Internal Covariate Shift [29] and Batch Normalization can play a crucial role on mitigating its effects. Consequently, it can actuate the gradient descent operation to a faster convergence, but it can also lead to overall higher accuracy and render the network more robust against overfitting.

#### 2.1.3. Prototypical CNN Architectures

A typical CNN architecture targeting the enhancement of remote sensing images is shown in Figure 1. In addition to the typical convolutional layers, start-of-the-art design also include architectural components like residual layers and inception modules which have been shown to significantly increase performance.

*Residual architectures*. An important extension of typical CNN architectures is the introduction of the residual-learning concept [30], first proposed for single-image super-resolution [31], which has allowed CNNs to grow much deeper without suffering the problem of vanishing/exploding gradients. In essence, instead of learning a direct map from low-quality inputs to high-quality outputs, the CNN is tasked with learning the residual, i.e., the difference between the low and high-quality signals, which typically represents missing high-frequency information, at least for the case of super-resolution. To allow networks to capture and extract features from multiple scale, *skip connections* between different layers have also been considered and are now part of state-of-the-art approaches.

*Inception architectures*. An important design choice for a CNN is the appropriate kernel size, where typical values are 3×3 and 5×5 for 2D image analysis. However, a single choice of kernel size might lead to suboptimal performance. To address this challenge, the inception architectures [32,33] introduce different size kernels at each convolutional layer and concatenate the filter outputs, while the importance of each kernel is automatically adjusted during the training process. This way, features of different spatial extend are automatically selected for each layer; however, this benefit come at a price of computational complexity due to the include in the number of trainable parameters.

*Hourglass architectures*. While in problems involving the classification of the input, the output space is extremely smaller compared to the input space, for image enhancement problems, the dimensions of the input and output spaces are typically the same, i.e., the size of the image itself. As such, CNN architectures which progressively reduce the inputs to subsequent layers through pooling layers are not appropriate. At the same time, critical features may reside at much higher dimensions compared to the ambient dimensions of the input/output. A representative example of this architecture is the SRCNN architecture by Dong et al. [15,34] which progressively shirks and then expands the feature extraction process through the introduction of deconvolution layers. A representative U-NET type architecture is shown in Figure 2.

*U-Net architectures*. Another very popular CNN architecture is the so-called U-Net architecture, initially proposed for the problem of semantic segmentation in medical images [35]. In the U-Net architecture, a symmetric design is followed where progressively convolutional and max pooling layers is introduced leading to a very compact representation of the input image (contracting path), which in sequence is expanded back to the dimensions of the input by convolution and upsampling operators (expansive path).

### 2.2. Autoencoders (AE)

A classical autoencoder (AE) is a deterministic DNN comprised of an input and an output layer of the same size with a hidden layer in between, which is trained with back propagation in a fully unsupervised manner, aiming to learn a faithful approximation of the input [36]. Specifically, the formulation considers both as an input and as the output examples $s\in R^{N}$, and encodes the information through a nonlinear function $\sigma :R^{N}\to R^{M}$, such that each input vector is mapped to a new feature space via *M* hidden units. An illustration of a single layer AE is shown in Figure 3

Formally, a single-layer AE network consists of the input layer units $s\in R^{N}$, the hidden layer unit, $h\in R^{M}$, and the output units $\hat{s}\in R^{N}$. and the objective is to learn a set of weights $W\in R^{M\times N}$, along with the associated encoding bias $b_{1}\in R^{M}$, in order to generate compact and descriptive features $h=\sigma (Ws+b_{1})$ that can accurately reconstruct the input example. A typical example of the function σ is the logistic sigmoid function. Afterwards, the decoding of h is performed using the weight matrix $V\in R^{N\times M}$ that connects the hidden layer with the output units $\hat{s}=\sigma \left(Vh+b_{2}\right)$, where $b_{2}\in R^{N}$ stands for the decoding bias. In most cases, tied weights are considered such that $W=V^{T}$. Stacked Autoencoder (SSE) is considered the deep-learning extension of AE where multiple shallow (single-layer) AE are stacked together and trained using greedy methods for each additional layer [37,38]. By applying a pooling operation after each layer, features of progressively larger input regions are essentially compressed into smaller ones, and thus can facilitate several classification or clustering tasks [39,40].

#### 2.2.1. Sparse Autoencoders

An AE is closely related to the Principal Component Analysis (PCA), by performing an internal dimensionality reduction, an over-complete nonlinear mapping of the input vector s can also be targeted by allowing more elements in the hidden layer, i.e., M>N. To avoid trivial solutions such as learning the identity function, additional constraints need to be imposed such as sparsity of the internal representation. Consequently, to learn representative features, the error of the loss function:(1)$$\begin{array}{c} L\left(s,\hat{s}\right)=\frac{1}{2}\sum_{j=1}^{N}\left|\right|{\hat{s}}_{j}-s_{j}{\left|\right|}_{2}^{2}, \end{array}$$
should be minimized, adhering to a sparsity constraint. In the aforementioned formulation, s and $\hat{s}$ correspond to the input and the output data, respectively. To restrict the average activation to a small desired value, the sparsity constraint is imposed by introducing a Kullback–Leibler divergence regularization term such that:(2)$$\begin{array}{c} L\left(s,\hat{s}\right)=\frac{1}{2}\sum_{j=1}^{N}\left|\right|{\hat{s}}_{j}-s_{j}{\left|\right|}_{2}^{2}+\beta \sum_{j=1}^{M}\mathrm{KL}(\rho ||\rho_{j}), \end{array}$$
where β is a sparsity regularization parameter, *M* is the number of features, ρ is the average activation of h, $\rho_{j}$ is the average activation of the $h_{j}$-th vector over the input *N*-data, and KL denotes the Kullback–Leibler divergence regularization term defined as:(3)$$\begin{array}{c} KL(\rho ||\rho_{j})=\rho \log\frac{\rho }{\rho_{j}}+\left(1-\rho \right)\log\frac{1-\rho }{1-\rho_{j}} \end{array}$$

Consequently, the network learns weights such that only a few hidden nodes are activated by the given input.

#### 2.2.2. Denoising Autoencoders

In a denoising autoencoder, the network is trained to reconstruct each data-point from a corrupted input version [41]. Similar to the sparse autoencoder case, during the training process, the main objective is to learn both the encoder and decoder parameters. For this purpose, a noise distribution $p(\tilde{x}|x,n)$ is defined, where the parameter *n* is associated with the amount of introduced noise. The autoencoder’s weights are trained to reconstruct a random input from the training distribution. Formally, this process is summarized as follows:(4)$$\begin{array}{c} \left(\theta^{*},\theta^{\prime *}\right)=\arg\min_{\theta ,\theta^{\prime }}E_{\left(X,\tilde{X}\right)}\left[L(X,g_{\theta^{\prime }}\left(f_{\theta }\left(\tilde{X}\right)\right)\right], \end{array}$$
where L stands for the loss function.

One of the most crucial parameters in the denoising autoencoder is the noise distribution *p*. Gaussian noise is a common choice for continuous x, defined as
(5)$$\begin{array}{c} p\left(\tilde{x}|x,n\right)=N\left(\tilde{x};x,n\right), \end{array}$$
while a masking noise can be used, when x is a binary variable, as:(6)$$\begin{array}{c} p\left(\tilde{x_{i}}|x_{i},n\right)=\{\begin{array}{cc} 0, & \mathrm{with}\;\mathrm{probability}\;\;n\\ x_{i}, & \mathrm{otherwise} \end{array}. \end{array}$$

In any case, the amount of noise *n* affects the degree of corruption of the input. If *n* is high, the inputs are more heavily corrupted during training. Additionally, the noise level provides a significant effect on the representations that are learned via the network. For instance, in scenarios where input data are images, masking only a small number of pixels will bias the process of learning the representation to confront only local corruptions. On the other hand, masking larger areas will enforce the network to use information from more distant regions.

#### 2.2.3. Variational Autoencoders

Presently, variational autoencoders (VAE) depict high performance in several generative modeling tasks, such as handwritten digit recognition, face recognition, physical modeling of natural scenes, and image segmentation among others [42]. Generative modeling focuses on learning models of distributions P(X), defined over data-points *X* that are spanned on a high-dimensional space X. However, the main limitation regarding the training phase of a generative model arises from the complicated dependencies among the model’s dimensions. To justify that the learned model represents efficiently the input dataset, we need to ensure that for every data-point *X* in the dataset, there are exist latent variables able to synthesize successfully the input data-points.

Formally, let z be the vector of latent variables lying on a high-dimensional space Z, which can be sampled according to some probability density function (PDF), P(z), defined over Z. Additionally, let f:Z×Θ→X, f(z;θ) be the family of deterministic functions parameterized by a vector θ in some space Θ. The goal is to optimize θ such that z can be sampled from P(z), and f(z;θ) can approach the input data-points of X with high probability. The aforementioned process is summarized as follows:(7)$$\begin{array}{c} P(X)=\int P(X|z;\theta )P(z)dz \end{array}$$

Therefore, the main task of variational autoencoders (VAE) is the maximization of the probability density function, P(X). Unlike traditional sparse or denoising autoencoders, variational autoencoders require no tuning parameters or sparsity penalties, while the output distribution is often chosen to be Gaussian, such as: $P\left(X|z;\theta \right)=N(X|f\left(z;\theta \right),\sigma^{2})$.

### 2.3. Generative Adversarial Networks

Generative Adversarial Networks (GANs) represent a radical new approach in DNN which have been recently presented in [43] and have led to significant increase in terms of performance (e.g., accuracy, quality), as well as have enabled the realization of new types of learning protocols, such as the synthesis of extremely realistic images [44]. In the context of the GAN framework, instead of a single DNN, training involves two DNNs, a “generator” and a “discriminator” network, where the former synthesizes realistic signals given an input, and the later classifies inputs as real or synthetic.

In the original formulation of the GAN framework [43], the generator is seeded with randomized noise input which leads to different output realizations, depending on its statistical characteristics. For image enhancement problems, a variant of GANs, called conditional GANs (cGANs), is more appropriate since the input to the generator is the image itself, although it could be very different from the output, such as an edge map [45]. A typical cGAN architecture is shown in Figure 4

The prototypical GAN architecture for inverse imaging problems, involves an iterative training protocol that alternates between the synthesizing of a high-quality image $I^{S}$ given a low-quality input image $I^{IN}$, performed by the generator *G*, and the classification of the high-quality image as real $I^{R}$ or synthetic $I^{S}$, executed by the discriminator *D*. Therefore, training a GAN corresponds to solving a min-max problem where the objective is to estimate the parameters (weights and biases) of the generator $\theta_{G}$ and the discriminator $\theta_{D}$, given by
(8)$$\min_{\theta_{G}}\;\max_{\theta_{G}}E_{I^{R}∼p_{train}\left(I^{R}\right)}\;\left[\log D_{\theta_{D}}\left(I^{R}\right)\right]+E_{I^{IN}∼p_{G}\left(I^{IN}\right)}\;\left[\log\left(1-D_{\theta_{D}}\left(G_{\theta_{G}}\left(I^{IN}\right)\right)\right)\right]$$

An important work that demonstrates the capabilities of GAN in typical inverse imaging problems, is the super-resolution GAN (SRGAN) architecture [46] which combines multiple loss functions, including adversarial and perceptual, for single-image super-resolution.

### 2.4. Performance Evaluation

Obtaining examples for training a DNN model is in general is very challenging process, which for the case of classification, requires significant labor and time. For the case of enhancement however, the process can be substantially simplified by employing the Wald’s protocol [47].

#### 2.4.1. Evaluation Metrics

Peak Signal-to-Noise Ratio (PSNR) is a well-established image quality metric, typically employed for 8-bit images, an expresses image quality in decibels (dB), so that higher is better.Mean Squared Error (MSE) is a generic signal quality metric where lower values are sought.Structural Similarity Index (SSIM) is an image quality metric which considers the characteristics of the human visual signal, such that values 1 indicate better performance.Signal-to-Reconstruction Error (SRE) is given in decibels (dB) and the higher values indicate higher quality.Spectral Angle Mapper (SAM) is an image metric typically employed for MS and HS imagery and values close to 0 indicate higher quality reconstruction.

#### 2.4.2. Remote Sensing EO Platforms

Sentinel 2 satellite carrying the 12 band MSI instrument with spatial resolution between 10 and 20 m for most bands in the visible, near and shortwave infrared.Landsat 7 and more recently Landsat 8 also acquire HS imagery over 11 bands in the visible to infrared range at 30 and 100 m resolution, respectively.SPOT 6 and 7 provide PAN imagery at 2 m spatial resolution and MS at 8 m resolution offering a swatch of 60 kmPleiades 1A and 1B acquire PAN imagery at 0.5 m and MS at 2 m, offering high agility for responsive tasking.QuickBird and IKONOS are two decommissioned satellites offering 0.6 m and 1 m panchromatic (PAN) and 2.4 and 4 m 4-band MS imaging, respectively.Worldview-3 acquiring 0.31 cm PAN and 1.24 m eight-band MS imagery, as well as shortwave infrared imagery at 3.7 m resolution.GaoFen-1 is equipped with two instrument sets, 2 cameras offering 2 m PAN, 8 m MS and four wide-field cameras acquiring MS imagery at 16 m resolution.Jilin-1 satellite family by Chang Guang Satellite Technology Co. Ltd, Changchun City, Jilin Province, China [48].The Airborne visible/infrared imaging spectrometer (AVIRIS) is a 224 band HS airborne instrument for which several annotated acquisitions are available.The Hyperion instrument aboard the EO-1 satellite is another well-known HS instrument capturing imagery over 200 bands, which was decommissioned in 2017.ROSIS is a 102 band HS airborne instrument which is also associated with annotated acquisitions.

#### 2.4.3. Typical Datasets and Data Sources

UC Merced contains 100 images of size 256×256 from 21 different classes at 0.3 m resolution.NWPU-RESIS45 [49] contains 700 images from 45 scene classes (31,500 images in total).RSCNN7 contains 2800 images from seven different classes.NWPU-RESISC45 is a data set of aerial imagery of scenes from 45 categories, and each category contains 700 images of size of 256×256 pixels.Indian Pines, Salinas, Cuprite and Kennedy Space Center from AVIRIS.Pavia Center and University from ROSIS offers 102 spectral bands over an urban area as 1.3 m resolution.Botswaba from EO-1 Hyperion offers 145 calibrated bands over different land cover types including seasonal swamps, occasional swamps, and drier woodlands.Kaggle Open Source Dataset and more specifically the Draper Satellite Image Chronology contains over 1000 high-resolution aerial photographs taken in southern California.Caltech, which is a publicly available aerial database that comprises an aircraft dataset and a vehicle dataset.

## 3. Super-Resolution

### 3.1. Super-Resolution of Remote Sensing Observations

The problem of super-resolution is among the most studied problems in the context of DNN for remote sensing observation enhancement, where the objective is to increase the spatial and/or spectral resolution of observations, either as an end goal, or as an intermediate, in an effort to achieve higher classification accuracy. To train DNN architectures, the typical approach involves the generation of synthetic data through the application of a forward downsampling process by a given factor, and then the evaluation of the performance on the inverse upsampling process following the Wald’s protocol [47]. Three of the most prolific CNN architectures, the Super-Resolution CNN (SRCNN) [34], the Very Deep Super-Resolution (VDSR) [31] and the Enhanced Deep Super-Resolution (EDSR) [50], have shown significant capacity in natural image super-resolution. In terms of methodological approach, the SRCNN is first CNN-based approach applied to this problem, the VDSR extends the SRCNN by introducing the notion of residual architectures and the EDSR introduces additional modifications making it among the highest performing methods for generic image super-resolution available today. We note that unlike typical CNN architectures, implementations that focus on super-resolution do not involve a pooling layer, since this operation reduces the spatial resolution, which is not desired in this context.

#### 3.1.1. Single-Image Super-Resolution

The use of DNN and CNN architectures for single remote sensing image super-resolution was first proposed by Liebel et al. [51]. In this work, the authors consider the paradigm of SRCNN [34] for the spatial resolution enhancement of Sentinel-2 observations, but focus only on observations from the third band (559 nm). The results demonstrate that using a pre-training SRNN network leads to significantly worse performance compared to “naive” methods such as bicubic interpolation, while fine-tuning the network to the particular type of observation, leads to a marginal increase in performance (0.3 dB). Similarly, in [52], the case of CNN for single-image super-resolution by factors of 2,3 and 4 is investigated using observations from the SPOT 6 and 7 and the Pleiades 1A and 1B satellites. The authors compare the VDSR and the SRCNN architectures and report significant benefits of the VSDR compared to the SRCNN; however the gains offered by the VDSR are marginal compared to bicubic interpolation (≤1 dB in PSNR). Huang et al. [53] also experimentally demonstrate that the VDSR applied to Sentinel-2, does not produce impressive results. Therefore, they introduce the Remote Sensing Deep Residual-Learning (RS-DRL) network, in which the features extracted by the SRCNN are introduced instead of the residual, achieving better performance than VDSR on Sentinel-2 images.

To encode information at different scales, such as those present in remote sensing imagery, Lei et al. [54] propose a modification of the VDSR architecture by directly propagating features from different convolutional layers to a “local-global” combination layer, the so-called “multifork” structure. The capabilities of the proposed LGCNet architecture are evaluated on RGB imagery from the UC Merced dataset and the results demonstrate that: (i) generic single-image super-resolution methods such as the LGCNET and other DL methods such as SRCNN can achieve higher quality estimation compared to bicubic for scale factor 3 on aerial imagery (1–2 dB) and (ii) the proposed LGCNet method does not achieves any noteworthy improvement compared to generic DL approaches such as [15,34]. The extraction of features from multiple scales is also addressed by Xu et al. [55], who propose the DMCN architecture, a symmetric *hourglass*-structured CNN with multiple skip connections, termed memory connections, for single-image super-resolution. The reported results demonstrate an increase of less than 0.5 dB and 0.02 in terms of PSNR and SSIM respectively on the NWPU-RESISC45, UC Merced and GaoFen1 datasets compared to the LGCNet [54] architecture. An interesting remark of this work is that the introduction of downsampling and corresponding upsampling layers lead to around 53% and 67% reduction in memory footprint and inference time, respectively.

Multiscale feature extraction is also discussed in [56], where a joint wavelet and CNN-based method is proposed for aerial image super-resolution. The method involves training multiple CNNs in multiple frequency bands generated by performing wavelet decomposition on aerial images, in an effort to approximate the wavelet multiscale representation and restore frequency features at different directions. For inference, each CNN is responsible for estimating the corresponding representation at the specific scale and the high-resolution image is obtained through wavelet synthesis. The evaluation results indicate that the proposed method achieves marginal improvements of around 0.1 dB compared to the VDSR architecture on noise-free images from the RSSCN7 dataset, while for the case of noisy imagery, a similar performance to the VDSR is reported.

Another super-resolution approach incorporating the wavelet transform and CNN equipped with local and global residual connections is proposed in Ma et al. [57] which addresses the issue by focusing on the frequency domain. Formally, the super-resolution CNN accepts as inputs four sub-band representations of the input image obtained by a 2D discrete wavelet transform and the original low-resolution input. The four input images are introduced to the proposed recursive residual CNN architecture and the outputs of the network are combined by an inverse discrete wavelet transform to generate the final high spatial resolution image. Evaluation of the method on the airplane images from the NWPU-RESISC45 dataset demonstrate marginal improvements (≤0.3 dB) compared to DRNN [58], a recently presented DNN scheme for natural image super-resolution. Extraction of features across multiple scales for single satellite image super-resolution is also explored in the work by Lu et al. [59] where a multiscale residual DNN is considered.

While moderate increase in the spatial resolution, e.g., ×2,×3, of single remote sensing RGB images has been to a large extend addressed, the case of higher scale factors, e.g., ×4,×8 is still extremely challenging. This challenge is picked up by the authors of the Fast Residual Dense BackProjection Network (FRDBPN) [60], which is heavily inspired by the method in [61], originally developed for natural color image super-resolution. The proposed methods employ upprojection and downscaling units, which consist of specific sequences of convolution and deconvolutions (dilated convolutions), as well as both global and local residual connections. Experimental results on the UCMerced dataset demonstrate a moderate (≤0.5 dB) increase in quality estimation compared to other DNN-based natural single-image super-resolution methods

The Generative Adversarial Network (GAN) framework, which involves the interaction between a generator network and a discriminator network to produce high quality and realistic reconstructions, has been recently considered for the super-resolution of remote sensing images. Haut et al. [62] focus on a generative model, proposing an *hourglass* architecture network which is trained in an unsupervised way, i.e., without using an external dataset of examples. Specifically, the generator network starts from random noise as input and iteratively refines the high-resolution image estimation by performing a downsampling step that generates a low spatial resolution data which is fed to the second upsampling step. Compared to a large class of unsupervised state-of-the-art methods, the proposed model achieves certain gains from the UC Merced, the RSCNN7 and the NWPU-RESIS45 datasets, especially for ×4 resolution enhancement, which is more than 0.5 dB for specific classes. Another GAN-based approach is also explored in [63], which involves a series of modifications to the prototypical SRGAN [46] architecture. A major novelty of this work is that training of the architectures is conducted under a transfer learning protocol using a dataset of natural images before fine-tuning the network on aerial images from the UC Merced dataset. The proposed approaches demonstrated an average increase of 0.5 dB compared to the generic SRGAN framework, which itself achieves a gain of 1.3 dB compared to the SRCNN framework.

Last, in [64], a GAN architecture is proposed which specifically targets the estimation of high-frequency edge information from remote sensing imagery for video acquiring platforms. The generator network is composed of two sub-networks, namely the ultradense subnetwork which aims at reconstructing a baseline high-resolution image and the edge-enhancement subnetwork which addresses the issue of noise contaminating the estimated edges. Comparisons with state-of-the-art super-resolution techniques for natural images demonstrate significant improvements, where for the Kaggle Open Source dataset, the proposed method surpasses SRCNN [34] by more than 2 dB, the VDSR [65] by 1.5 dB and the GAN-based SRGAN [46] method by 1 dB.

#### 3.1.2. Multispectral and Hyperspectral Image Super-Resolution

In addition to single color image super-resolution, approaches for the enhancement of both spatial and spectral resolution of MS and HS observations have also been proposed. Efforts such as the ones by Hu et al. [66] as well as by C. Wang et al. [67] were among the first to consider DNNs for HS super-resolution; however, their investigation focused on natural scene, not remote sensing ones. One of the first DNN-based approaches for remote sensing HS super-resolution was proposed by Yuan et al. [68] where the authors use the single-image super-resolution SRCNN architecture, pre-trained on natural images, for enhancing each band independently, and then employ a collaborative non-negative matrix factorization in order to enforce spectral consistency through the encoding of end-member information. In [69], another CNN architecture similar to the SRCNN [34] is considered for the automated spatial resolution enhancement of low-resolution MS observation, exploiting overlapping observation between two 4-band sensors aboard different satellites, for generating training examples. The authors specifically consider the fusion of low-spatial-resolution images from the 740 km swath Advanced Wide-Field Sensor sensors and high-spatial-resolution images from the 140 km swath Linear Imaging Self Scanner sensors aboard the Indian Space Research Organisation’s Resourcesat 1 and 2.

A two-step approach is considered in [70], where spatial resolution enhancement is achieved through an appropriate combination of deep Laplacian Pyramid Networks (LPNs) [71] and spectral enhancement is addressed through a dictionary learning with non-negativity constraint approach. Specifically, the LPN is trained using pairs of low- and high-resolution natural imagery and then applied for the super-resolution of each band individually. Then a dictionary encoding the spectral content, trained on low spatial resolution HS observations, is employed for increasing the spectral resolution of the estimated bands. Experimental results with observations from the Indian Pines and the Pavia Center dataset demonstrate substantial gains, 3.4 dB and 2.2 dB respectively, compared to the SRCNN [34] method which achieved the second-best performance.

An issue related to the previous CNN-based super-resolution methods is that quality, and therefore the loss function, is quantified through the $\ell_{2}$ norm, i.e., $\sum_{i}{\left(x_{i}-{\hat{x}}_{i}\right)}^{2}$, where *i* indexes the examples. Although this loss function may be appropriate for single-band imagery, for the MS/HS case, it can lead to spectral inconsistencies. To address this challenge, Zheng et al. [72] proposed a multi-losses function Network for simultaneously enforcing spatial and spectral consistency. The composite loss function in this case measures the normalized $\ell_{1}$ norm between bands for capturing spatial quality and the cosine plus the $\ell_{1}$ between spectral pixels for spectral quality. Evaluation on the Pavia Center dataset demonstrated a substantial increase of 1.8 dB compared to the VDSR [31] and of 0.5 dB compared to EDSR [50], single-image/band approaches.

A specific situation of MS super-resolution, which is however of great importance, involves increasing the resolution of observation from the ESA Sentinel 2 satellite. A CNN-based super-resolution approach, inspired by the state-of-the-art single natural image EDSR method [50], is considered for Sentinel 2 imagery [73], targeting the estimation of 10 m ground sampling distance images from 20 m and 60 m images. Training examples are generated through a simulated downsampling processing of the available observations, based on the assumption that this degradation process is closely related to the actual reduction in the observed spatial resolution. An important observation, made by the authors, is that the proposed CNN can be generalize to any location over the Earth, even though it is trained using examples from a small number of representative regions. The results indicated that a very deep CNN architecture can achieve significantly higher quality estimation, in the order of 6 dB for the SRE metric compared to competing methods.

The problem of Sentinel 2 super-resolution is also investigated in [74], where a residual-structured CNN is considered. A detailed investigation of the impact of different hyper-parameters on the performance of observation enhancement from two locations, an urban and a coastal scene, revealed that: (i) increasing the number of training patches can lead to worse performance due to overfitting for certain scenes, (ii) increasing the number of training epoch offers minimal improvement, (iii) the number of residual blocks is not a critical design parameter, and (iv) optimal values exist related to patch size, and deviations from this size lead to worse performance. Gargiulo et al. [75] also consider Sentinel 2 observations but focus on the super-resolution of the 11^*th*^ SWIR band by posing the problem as a pan-sharpening one and employing a CNN architecture similar to [76].

A limitation of typical DNN based on SR methods applied to HS observations is that when 2D convolutions are applied, the convolutional layers will tend to fuse information across spectral bands. For many cases, different bands are characterized by different SNR, therefore the fusion operation can introduce noise to the extracted features. To address this issue, a Separable-Spectral convolution and Inception Network (SSIN) is proposed in [77] to both extract band-specific features and fuse features across bands. Validation of the Pavia center dataset demonstrated in increase of 0.8 dB compared to the VDSR [31] method for upscaling factor ×2 but almost no gains for ×4.

The *simultaneous* encoding of both spatial and spectral information for the purpose of satellite HS super-resolution was first proposed in [78]. To achieve this objective, a 3D CNN, i.e., a CNN with three-dimensional convolutional kernels, is employed. The proposed architecture assumes an interpolated image as input and the application of sequences of 3D convolution for estimating the high-quality output, measured in terms of MSE. Experimental validation using observations from the ROSIS and HYDICE sensors demonstrate that using 3D convolutions lead to higher quality compared to 2D methods such as [51] while another observation is that the optimal performance is achieved by a moderate number of convolutional layers and that more layers lead to marginal decrease in performance.

The potential of simultaneous spatio-spectral super-resolution in addition to sub-pixel classification of MS observations from aerial vehicles (drones) is examined in [79]. To achieve both tasks, a network inversion-based architecture is proposed where a CNN architecture accepts both the low-resolution image, as well as image-derived features such as histograms-of-oriented-gradients. Experiments on airborne datasets such as Indian Pines and Salinas, indicate that this super-resolution approach better preserves the spectral fidelity when compared to sparse coding approaches which generally in corrupted dictionaries and sparse codes.

Ran et al. [80] consider a hierarchical CNN architecture which learns an end-to-end mapping from low-resolution to high-resolution images, by combining feature extraction and edge enhancement in the hierarchical layers. Extensions of this approach based on residual learning and multiscale version are also investigated for further improvements. The performance of the proposed approach is compared with bicubic interpolation and dictionary-based methods on the areal color, MS and HS images suggest that better recovery quality and reduced computational requirements compared to dictionary-based methods.

Last, we note two interesting approaches for spectral super-resolution using CNN. The first one explores the use of various CNN architecture for estimating extended spectral resolution MS imagery from 3 channel RGB images, without resorting to additional hardware components or imaging protocols, by exploiting the correlations between the RGB values and the corresponding HS radiance values. The HSCNN+ method [81] proposes deep residual and densely connected CNN architectures, which do not require an explicit upsampling pre-processing operator. The proposed method achieved the first place in the NTIRE 2018 Spectral Reconstruction Challenge for both the “Clean” and “Real-World” tracks, which however consider natural scenes, not remote sensing ones. Similar approaches are also considered in [82,83].

The second approach involves the use of DNN in the framework of Compressed Sensing [84]. Unlike traditional MS/HS architecture, approaches based on Compressed Sensing employ randomized sampling processes to acquire some form of compressed observations and subsequently employ optimization algorithms for estimating the image. By doing so, such approaches are able to overcome different trade-offs related to acquisition, including producing imagery with higher spatial resolution compared to the number of detectors used or provide a full spectral resolution MS from a single observation e.g., [85,86]. In the context of Compressed Sensing, Yuann et al. [87] consider the super-resolution of observations acquired by a Compressed Sensing architectures, specifically the CASSI [88], using CNN architectures, which allow demonstrates promising performance under simulated conditions. A similar approach involving the recovery of HS observations from Compressed Sensing measurements is also considered in [89], while in [90], in addition to the recovery process, optimization of the acquisition process is also explored.

#### 3.1.3. Video Super-Resolution

While most remote sensing platforms acquire still imagery, grayscale (panchromatic), RGB, or hyperspectral, a new breed of platforms focuses on the acquisition of video imagery, including the Jilin-1 satellites. To achieve however high temporal resolution, the imaging architecture typically needs to sacrifice spatial resolution, mandating the application of super-resolution for providing high-quality spatio-temporal observations. In [91], a CNN architecture based on the VDSR [31] is explored for super-resolution of video satellite frames. Training examples from a high-spatial-resolution static imaging satellite (Gaofen 2) are used, while the learning process is applied for super-resolving single frames from video sequences, acquired by another platform (Jilin 1). To address issues related to inability of existing methods to enhance image boundaries, the authors proposed an appropriate padding of the boundaries. The proposed method achieves a gain of 0.5 dB when trained on high-resolution satellite imagery compared to VDSR trained on natural imagery.

Another approach for satellite video super-resolution is proposed in [92] in which extracted video frames are used for both training and testing the DNN architecture. The proposed network is a variant of the SRCNN method [34] where a deconvolution layer is introduced for producing high-resolution outputs, thus removing the need for pre-processing. Validation for an upscale factor of 3 on a diverse set of scenes acquired by the Jilin-1 platform demonstrates an increase between 0.5 and 1.7 dB compared to SRCNN. Jiang et al. [93] consider a two-stage process for video frame super-resolution, where a pre-trained CNN with an architecture similar to [16] produced multiple features at the first stage, which are then combined though a densely connected network at the second stage. The proposed PECNN architecture achieve a minor increase in terms of no-reference quality metrics, such as the average gradient of the naturalness image quality when evaluated on aerial imagery from the Kaggle dataset; however, the method demonstrates a substantial reduction in terms of processing time per image.

Satellite video super-resolution is also explored in [94] where a CNN variant termed deep distillation recursive network (DDRN) is introduced, featuring (i) groups of ultradense residual blocks encoding multiple short and long skip connections without substantially increasing the computation and memory requirements, (ii) a multiscale purification unit for the recovery of high-frequency components and (iii) a reconstruction module which corresponds to a shuffling operator for producing the final high-resolution image. Evaluation on observations from Jilin-1 satellite, as well as from the aerial imagery available through the Kaggle Open Source Dataset, demonstrates a gain of more than 1 dB compared to the VDSR [31] method for an upscale factor of 4 when trained on the same dataset.

### 3.2. Discussion

The number of applications of DNN in remote sensing super-resolution has increased dramatically in the past 2-3 years, demonstrating superior performance compared to other state-of-the-art method. In Table 1 we provide a high-level grouping of the different methods discussed above.

In addition to the qualitative grouping of the different super-resolution approaches presented in Table 1, Table 2 and Table 3 provide some quantitative performance results for single-band and multi-band image super-resolution respectively. In both cases, we report the gains offers by each method compared to the bicubic interpolation measured in terms of PSNR, for different requested super-resolution factors. Regarding the case of single-band images and the case of ×4 super-resolution, the reported results indicate that the use of GAN-based architectures, pre-trained on generic imagery, lead to the higher increase in terms of visual quality. For the case of multispectral imagery, the increase in performance is significantly lower, which the results suggest that exploiting both spatial and spectral information lead to the best performance.

Based on our analysis of the current state-of-the-art, several observations can be made regarding the problem of remote sensing super-resolution.

In terms of single-image, single-band super-resolution, approaches based on the GAN framework appear to achieve the highest quality image estimation. This observation for the case of remote sensing is in-line with results for methods applied in generic natural image.For the case of spatial-spectral super-resolution, the best performing approaches pay special attention of simultaneously increasing the spatial resolution without introducing unwanted artifacts in the spectral domain, unlike early approaches which consider each spectral channel independently.Exploiting temporal information, which for many remote sensing cases, can be directly translated to access to multi-view imagery given the regular imaging protocols in satellites imaging.Super-resolution of video sequences require the exploitation of both spatial and temporal information, such as the approach proposed in [95] which encodes spatio-temporal information using a recurrent residual network and is applied to the restoration of generic video sequences. This methodology however has not been explored in the context of remote sensing, where methods primarily consider single frames.Although numerous approaches have been proposed for addressing the combination of two resolution dimensions, methods which can simultaneously enhance remote sensing images along their spatial, spectral and temporal dimensions have not been considered.

The above observation can be used to outline an exemplary architecture of MS/HS super-resolution shown in Figure 5. This exemplary architecture captures the essence of the most successful paradigms in the area, using 3D convolutions, skip connections over multiple layers, dilated convolution for upsampling and the use of multi-objective loss functions.

## 4. Pan-Sharpening

While both high spatial and spectral resolution observations are desired in remote sensing, imaging systems must sacrifice the resolution in one domain to achieve the requested performance in the other. As a result, the majority of spaceborne remote sensing imaging systems acquire two types of observations, high spatial resolution single-band panchromatic (PAN) images, or moderate to high spectral resolution multispectral (MS) or hyperspectral (HS) observations at much lower spatial resolution. For instance, the SPOT 6 and 7 satellites acquire PAN imagery at 1.5 m spatial resolution and 6 m spatial resolution for the 4 band MS, while the WorldView-4 acquires PAN imagery at 0.31 m and 4 bands MS at 1.24 m. Pan-sharpening methods explore how the two types of observations can be combined in order to produce high spatio-spectral observations [96]. For this problem, several DNN approaches have been recently proposed.

### 4.1. AE-Based Approaches

Huang et al. [97] are among the first to consider DNNs for the problem of pan-sharpening remote sensing observation. Specifically, they propose a modified Sparse Denoising Autoencoder (MSDA) architecture, aiming at learning the patch-level relationship between low and high spatial resolution MS and PAN images. The authors consider a series of pre-trained MSDAs which form a stacked MSDA, and subsequently fine-tune the entire network to infer high spatial resolution image patches for each MS band, which are then appropriately averaged to produce the final image/cube. The method is evaluated on two different satellite datasets from the IKONOS and QuickBird satellites, obtaining higher quality imagery compared to conventional algorithms.

To extract the most representative features from each source, a multi-level neural network approach for the Pan-sharpening problem is proposed in [98]. The authors propose a network composed of four individual Stacked Autoencoders (SAE) networks, and specifically a variation called Denoising SAE, which are coupled together to derive the desired high-resolution MS image. Concerning the role that each of the separate network play, two of them represent the high-resolution PAN image and the upsampled MS image in a new feature space via SAE, while the subsequent third one builds the connection (i.e., mapping) between these two representations. Finally, a fine-tuning fourth network is introduced at the highest level to ameliorate the learned parameters of the networks via joint optimization through backpropagation. The proposed method is evaluated on the QuickBird dataset, and managed to outperform not only conventional pan-sharpening techniques but deep learning based ones (i.e., [99]) as well.

A multi-resolution analysis framework for confronting the Pan-sharpening problem is proposed in [100] where a two-stage DNN scheme based on the SAE framework is investigated. In the first stage a low-resolution PAN image is constructed by its high-resolution counterpart via the help of two autoencoders (the second one is trained with input the features derived by the first one). In the second stage the desired high-resolution MS image is reconstructed via its low-resolution version according to the respective relationship between PAN images. Once both AE are trained alone, the whole network (i.e., both together) is fine-tuned to optimize its hyper-parameters. The proposed approach is tested in two separate datasets (namely QuickBird and WorldView-3), where it outperformed its competitive algorithms. A possible shortcoming of the method could be considered the absence of comparison with other deep learning-based methods, which seem to be state-of-the-art in the field in recent years.

A recently approach using AE for pan-sharpening is proposed by Azarang et al. [101] where the author consider a convolutional AE architecture, composed of an encoding and the associated decoding stage from improving the spatial information of the low-resolution MS bands. The objective is achieved by learning the nonlinear relationship between a PAN image and its spatially degraded version at a patch level, and uses the trained model to increase the spatial resolution of each MS band independently. Evaluation on data from QuickBird, Pleiades and GeoEye-1 demonstrate some gains compared to other non-DL-based methods.

### 4.2. CNN-Based Approaches

In addition to AE, CNNs have also been considered for pan-sharpening. One of the first attempts to employ CNNs in pan-sharpening is reported in [102] where the authors consider a CNN architecture based on the SRCNN [34] architecture for increasing the spatial resolution of MS observations in a band-by-band fashion and then use the spatially enhanced band corresponding to the PAN image for adjusting the original PAN image before subsequently applying a Gram–Schmidt transform to fuse the MS with the PAN observations. Training of the model is performed on natural images from the 2013 version of ImageNet dataset while for the validation, observations from QuickBird satellite are employed, demonstrating higher quality estimation compared to competing non-DL methods.

Another early attempt in using CNN for pan-sharpening is explored in [76] which also borrows ideas from the SRCNN super-resolution architecture and extends them to tackle the pan-sharpening problem. The key novelty of this work involves the upsampling of the low spatial resolution 4 band MS observations to the resolution of the PAN image and stacking both observations into a 5 component input. The authors further propose a nonlinear radiometric parameter (typical in remote sensing applications) injection as a surplus input to the network, which also contributes to an increase in the output image quality. The performance of the proposed approach, termed PNN, is validated using 3 different datasets from the IKONOS, GeoEye-1, and WorldView 2 satellites, where it outperforms the competitive non-DL algorithms with respect to various full-reference and no-reference metrics.

Another early attempt in the application of CNNs for par-sharpening is proposed by Li et al. [103] who, inspired by the VDSR architecture [31], propose the introduction of convolution/deconvolution layers and residual connections. The proposed method, called DCNN, is capable of estimating the full resolution output, not by averaging different patches but n a band/channel-wise fashion, achieving promising results with respect to the competing methods in two different datasets from the QuickBird and GeoEye-1 satellites.

Wei et al. [99,104] also propose a residual CNN architecture, to evaluate the performance of deeper networks as opposed to shallow architectures. To overcome the problem of vanishing gradient characterizing very deep architectures, the proposed approach, called DRPNN, considers a residual architecture with skip connections, inspired by the VDSR method for super-resolution, allowing the network to accept low-resolution images in the input, i.e., without the need for a upsampling pre-processing step. The method is tested in two separate datasets, namely QuickBird and WorldView-2, and demonstrates significant gains compared to traditional non-DL approaches, as well as well as a CNN one [76]. A similar residual CNN approach is also considered in [105] which is shown to outperform conventional algorithms, as well as the CNN-based method [76] in terms of performance on observations from the LANDSAT 7 ETM+.

In a similar vein, i.e., using residual connections for training deeper CNNs, the authors in [106] consider an architecture based on the ResNet [30] architecture. The proposed scheme incorporates domain specific knowledge to preserve both spatial and spectral information, by adding upsampled MS images to the output and training the network in the high-frequency domain. Experimental results demonstrate the superiority of the proposed method compared to conventional pan-sharpening methods as well as CNN-based [76]) in the WorldView-3 dataset, as well as greater generalization capabilities in new datasets, namely WorldView-2 and WorldView-3, where the proposed approach is capable of producing high-quality results without the need to be retrained.

Another CNN approach for pan-sharpening is proposed by [107,108] where the authors introduce a multiscale feature extraction by considering three different size convolutional filter at each layer to build a respective multiscale network, inspired by the inception architecture [33]. By concatenating the derived features across the spectral dimension, they retained as much spatial information as possible, while minimizing spectral distortion. To build a deeper network architecture, they adopt residual connection approach leading to sparse and informative features. The proposed architecture is tested in two different datasets from the QuickBird and WorldView-2 satellites, where it outperformed competitive approaches, including the CNN baseline [76], by 0.7 and 0.3 dB, respectively.

Building upon the method proposed in [76], which became a baseline for most of the aforementioned deep learning-based approaches for Pan-sharpening, the authors propose an updated approach in [109]. More specifically, they modified their initial approach by taking into account research that had taken place after their initial publication, by adopting the following changes: (i) use of $\ell_{1}$ loss function-instead of $\ell_{2}$; (ii) processing and learning in the image residual domain-rather than in the raw image one; (iii) use of deeper network architectures (in the [76], the network consisted of only three convolutional layers). The aforementioned modifications are tested in the GeoEye-1 dataset and compared to the initial setup proposed in [76], and proved to be quite beneficial in terms of performance as well as training time reduction.

Capitalizing on these observations, the authors extended the approach in [110] by making it adaptive to every new image that needed to be pan-sharpened via the trained network. For doing so, every new image fed a fine-tuning process on the pre-trained deep neural network (until convergence), and right afterwards it entered the trained model to be pan-sharpened at its output. To prove the validity of their -updated- approach, the authors tested it in four different datasets, namely IKONOS, GeoEye-1, WorldView-2 and WorldView-3, and compared it with their initial approach ([76]) as well as with conventional pan-sharpening techniques. The reported results in all datasets verified the merits of the new approach both in terms of performance metrics as well as of visual inspection.

Another approach involving CNNs for pan-sharpening is the RSIFNN architecture [111] which poses the problem as a fusion of PAN and MS inputs. Specifically, the RSIFNN introduces a two-stream fusion paradigm such that features from the PAN and MS inputs are first independently extracted and subsequently fused in order to produce high spatio-spectral observations, while a residual connection between the low-resolution MS and the outputs is also introduced. The authors considered PAN and MS (4 bands) observations the from QuickBird and Gaofen-1 satellites and generate synthetic examples through appropriate downsampling (Wald’s protocol) for comparing the performance of the proposed and various state-of-the-art methods, including the CNN-based method by Massi et al. [76] as well as the SRCNN method. The authors report a gain of around 0.3 dB compared to [76] and significant gains using additional metrics likes SAM, while also demonstrating a dramatic decrease (×6) in running time for inference compared to other CNN-based approaches. In a similar spirit, Zhang et al. [112] consider two-stream bidirectional Pyramid networks which can extract and infuse spatial information to MS from PAN imagery along multiple scales. Validation using GF2, IKONOS, QuickBird, and WorldView3 observations demonstrate some minor improvements in performance.

Wei Yao et al. [113] propose a pixel-level regression for encoding the relationship between the pixel values in the low- and high-resolution observation. For that case, they employed a CNN inspired by the U-Net architecture [35], but refined with respect to the problem at hand and trained with image patches. The proposed method obtained quite improved performance with respect to localization accuracy as well as feature semantic levels, compared to other DNN-based methods. At the same time, the proposed approach outperforms other sophisticated Pan-sharpening methods in two different remote sensing datasets in various indices of quality, leading to sharpened visual results both in terms of spatial details as well as spectral characteristics.

Increasing the spatial resolution of MS observation using PAN imagery using CNNs is also explored in [114] where a novel loss function penalizing the discrepancy in the spectral domain is proposed. The proposed scheme employs two networks, the first one called ’Fusion Network’ which introduces high spatial resolution information to upsampled MS images and the second one, the ’Spectral compensation’ network which tries to minimize spectral distortions. Validation on observations from Pleiades, Worldview-2 and GeoEye-1 demonstrate the proposed scheme achieves superior performance compared to [76,99] in almost all cases.

In the recently proposed method by Guo et al. [115], the authors use the latest CNN design innovations including single and multiscale dilated convolutions, weight regularization (weight decay) and residual connections for pan-sharpening Worldview-2 and IKONOS imagery. Compared to state-of-the-art methods, including [76,106], the proposed method achieved both superior performance in terms of quality (more than 0.5 in SAM compared to competing CNN approaches) and computational requirements for inference.

Building on the disruptive GAN framework, the authors in [116] proposed a two-stream CNN architecture (one stream for the PAN images and the other one for the MS ones) as the generator of high-quality pan-sharpened images, accompanied by a fully convolutional discriminator. In this way, the fusion of images is performed in the feature level (of the generator), rather than in pixel-level, reducing in this way the spectral distortion. The aforementioned approach is tested on two different datasets (namely QuickBird and GaoFen-1), where it outperformed conventional and deep learning-based ([76]) pan-sharpening methods, even if only the generator component is used solely. A similar idea is also explored in [117] where additional contains related to spectral features are imposed.

### 4.3. Discussion

A quick overview of the major approaches related to pan-sharpening remote sensing observation using DNN methods is presented in Table 4. Since in all cases the input is a PAN and an MS image, the table reports the different approaches groups along the specific DNN methodology employed. To offer some insights into the performance of each method, Table 5 report the gains of each method, measured in terms of Spectral Angle Mapper (SAM), compared a state-of-the-art non-DNN method of MTF-GLP [118] and one the first DNN-based approach, the PNN [76]. The results suggest that currently, the highest achieving DNN method is [115], based on its performance on pan-sharpening WorldView-2 observations. This observation suggests that using all available machinery during the realization of a CNN architecture offers significant performance gains. Based on the analysis of existing literature, several observations can be made

The majority of state-of-the-art performing methods rely on some variant of CNNs, typically employing fully convolutional architectures, while AE approaches are significantly less used for this problem.The introduction of successful CNN components such as inception modules and residual/skip connections lead to higher quality outputs.Among the best performing methods are the ones that employ two-stream architectures in which cases the PAN and MS are first analyzed independently by two separate paths and the extracted features are then jointly processed to produce the output.GANs has also been introduced in the problem of pan-sharpening with some promising initial results; however, more detailed analysis is required to justify any potential benefits.

Given the analysis of existing methods, Figure 6 presents a graphical illustration of an exemplary GAN architecture for pan-sharpening employing CNNs for both the generator and the discriminator network. This exemplary architecture captures the latest and most successfully architecture design considerations including the notion of skip connections, the use of upsampling layers using dilated convolutions, all the context of a GAN paradigm which has been shown to lead to the highest quality estimations.

## 5. Restoration

### 5.1. Denoising and Deblurring

Although MS and HS observations collected from remote sensing platforms provide valuable information, they are often inevitably contaminated with noise injected during acquisition and processing such as compression, which can jeopardize the performance of subsequent analysis. Observation denoising is thus a crucial process which has been extensively studied by the remote sensing community [119], while in recent years, several DNN architectures have been proposed for observation denoising.

In [120], the authors introduce a CNN with trainable nonlinear activation functions for HS denoising, where both the convolutional kernels’ weights as well as the nonlinear functions are learned from overlapping clean-noisy HS patch pairs serving as training examples. In this work, principal component analysis is initially applied to extract the major components from the highly correlated spectra, which are subsequently used for denoising the remaining components. Then, the major components and the remaining denoised components are inversely transformed to obtain a denoised HS. Compared with the state-of-the-art HS denoising methods, the experimental results on both synthetic and real remote sensing observations from the AVIRIS sensor (Salinas valley) demonstrate that the proposed method can achieve faster and more effective performance, at the cost; however, of not preserving the spectral information.

A HS denoising method which can properly remove spatial noise while accurately preserving the spectral information is proposed in [121], where a residual CNN with batch normalization is employed for learning the appropriate mapping between the noisy and the clean HS observations. In the denoising stage of the method, the learned mapping is adopted to reconstruct the clean spectral difference. Meanwhile, a key band is selected based on a principal component transformation matrix and denoised, acting as a starting point for the correction of other spectral bands. Comparative analyses on observations acquired by the ROSIS and the Moon Mineralogy Mapper (M3) acquired over the Aristarchus Plateau demonstrated that the method achieved higher performance compared to non-DL method, due to the fact that textures are retained in both large and small scales. The authors in [122] further extend the idea of denoising key bands similar to [121] while also using trainable non-linearity functions introduced in [120] for denoising the remaining non-key bands. Experiments on HYDICE imagery demonstrate that this model enjoys superior performance in spatial recovery and spectral preservation compared to other non-DNN approaches.

Unlike traditional CNNs that are limited by the requirements of the training sample size, Chen et al. in [123] introduce an efficient deep CNN model for the denoising of aerial images, which is appropriate for small training datasets. The learning algorithm of this model employs a multiscale residual-learning approach with batch normalization and dropout, which not only speeds up the training process but also improves the denoising performance of the model. Experimental results on images of the Caltech database showed that the proposed method achieves better performance compared to state-of-the-art denoising methods using small training datasets, while the comparison also included deep learning-based methods such as PNN [76] and SRCNN [34], which however, are not designed for image denoising.

Another DNN method for HS denoising is proposed in [124], by learning a nonlinear end-to-end mapping between the noisy and clean signals with a deep spatial-spectral CNN. More specifically, a 2D-CNN is used to enhance the feature extraction ability of single bands, and a 3D-CNN is employed to simultaneously use the correlation between adjacent bands, offering greater flexibility with respect to the available bands. To capture both the multiscale spatial and spectral features which characterize remote sensing imagery, different convolutional kernel sizes are employed. Experimental results from the Indian Pines and Pavia University datasets demonstrate that this method outperforms many of the mainstream non-DNN methods for HS denoising. A 3D CNN-based architecture is also explored in [125] for HS image denoising where the notion of atrous convolution is considered, a term which has being replaced by dilated convolution, which is able to expand the receptive field associated with each convolution kernel without increasing the number of parameters. The atrous convolutions, along with the introduction of skip connections and multiscale inception-like filters, demonstrated an increase in quality, in the order to 0.5 dB for some cases, compared to other non-DNN approaches.

While the previously reported approaches consider only one type of noise, typically Gaussian, in many real-life situations, especially in the case of HS imaging, one must deal with multiple types of noise, i.e., hybrid noise. The use of DNN for addressing hybrid noise in HS is considered by Zhang et al. [126] by simultaneously exploiting both spatial and spectral information through learning of spatial and spectral gradients at multiple scales. The proposed SSGN architecture further uses a spatial-spectral loss function for reducing distortions. Experiments performed on HS imagery from HYDICE, AVIRIS and Hyperion EO-1, confirmed that this method performs better at mixed noise cases (Gaussian noise + stripe noise + deal lines) than state-of-the-art non-DNN denoising algorithms. Addressing the removal of mixed noise types in HS is also considered by Chang et al. [127] where the proposed HSI-DeNet is a GAN-based approach which directly operates on the 3D data and incorporates high performance machinery such as residual learning and dilated convolutions. Experiments on HYDICE images indicate that this method outperforms the state-of-the-art in terms of both speed and performance. However, larger training datasets are needed to improve the generalization of this model to accommodate all kinds of complex noise categories and HS data.

A GAN-based methodology is also explored in [128] to deblur degraded remote sensing images in the context of image restoration. A common approach to solve this problem is the incorporation of various priors into a restoration procedure as constrained conditions that often lead to inaccurate results. In contrast, in [128] an end-to-end kernel-free blind deblurring learning method is presented that does not need any prior assumptions for the blurs. During training, a discriminator network is defined, with a gradient penalty which is shown to be robust to the choice of generator architecture, and a perceptual loss which is a simple $\ell_{2}$ loss based on the difference of the generated and target image CNN feature maps that focuses on restoring general content instead of texture details. This method can handle blur caused by camera shake and object movement, using fewer parameters compared to multiscale CNN. The experimental results on the Google map dataset showed that this method can obtain competitive results compared to other state-of-the-art methods. However, checkerboard artifacts still exist in some deblurred images, produced by the deconvolution-transposed convolution operation.

Finally, in [129], a novel deep memory-connected DNN with a large receptive field is introduced for remote sensing image restoration. Inspired by neuroscience concepts, the authors therein propose local and global memory connections to combine image detail information in lower layers with global information in higher layers of the network. In addition, to alleviate the computational burden and accelerate the training process, downsampling units are proposed to shrink the spatial size of the feature map. Experiments on three remote sensing datasets, namely from UC Merced, NWPU-RESISC45 and GaoFen-1, indicate that this method yields promising improvements and better visual performance over the current state-of-the-art, achieving high-quality image recovery for both known and unknown noise level at the same time.

### 5.2. Missing Data Recovery

Remote sensing observations are often hindered by missing measurements, which is attributed to poor atmospheric conditions, such as thick clouds, as well as internal satellite sensors malfunction such as dead pixels and scan lines. Thus, an issue of major importance is the estimation of the missing observations, also known as gap filling when considering time series of observations, by properly exploiting the spatial, temporal and spectral information [130]. Most of the earlier techniques (such as interpolation) deal with each domain separately, losing in that way crucial information, as most of them take into account linear correlations between the data. On the other hand, the recent breakthrough of the deep CNNs methods gives researchers the opportunity not only to reconstruct better the missing information but also incorporates knowledge from all aspects to address this challenge.

In [131] a CNN-based approach which exploits information under a unified spatial–temporal–spectral model is presented for missing measurements recovery, focusing on three representative cases, namely removing dead lines from the 6th band of Aqua MODIS instrument, scan lines in the Landsat Enhanced Thematic Mapper Plus and thick clouds. The proposed architecture considers two sources of data, i.e., spatial data with missing values and auxiliary spectral or temporal data without missing values, where initial features are independently extracted and subsequently fused in deeper layers of the CNN. The proposed CNN is augmented with several enhancements including multiscale feature extraction using dilated convolutions and residual learning using skip connections. Experimental results under various conditions using both simulated and real, as well as additional sources of noise such as registration error, demonstrate the very high performance and robustness of the proposed method.

A critical aspect of missing observation recovery, especially when temporal analysis is considered, is the preservation of causality. In their work, Das et al. [132] consider a deep learning-based approach for reconstructing missing time series data, which relies on making use of observations from both preceding and subsequent time instances while at the same time maintaining the causality constraint. The proposed network architecture comprises of four different modules, namely feature preparation, feature learning, prediction and data tuning, and specifically, which are organized in an ensemble of forward prediction DNNs based on the Deep-STEP architecture [133], each of which aims at predicting the immediate next missing observation in a sequence given previous time instances. Experiments performed using products (NDVI) derived from the Landsat-7 TM-5 satellite over the Barddhaman district, West Beghal-India, show clear superiority of the proposed approach over six competing algorithms under several quantitative metrics (i.e., MAE, RMSE, PSNR and SSIM).

A GAN-based method for estimating arbitrary shaped occluded regions of sea surface temperature images by exploiting historical observations is proposed in [134]. The authors consider the deep convolutional generative adversarial network [135] in order to address the unavailability of observations due to cloud occlusion, where the generator network is tasked with producing realistic estimation while the discriminator network must classify the inputs as real or synthesized. Estimating of missing regions is subsequently carried out by trying to estimating the closest vector representation of the uncorrupted image through the minimization of a loss function encapsulating the $\ell_{2}$ norm on the available observation, the adversarial error and the deviation of an average values. The results demonstrate a significant gain offered by the proposed approach compared to competing methods including the deep-learning method in [136] which is designed for generic imagery.

Conditional GANs are also considered in [137] for estimation of missing multispectral imagery by combining optical and radar multitemporal observations. The proposed methodology involves introducing pairs of registered optical and SAR patches from multiple dates for training the generation which is then tasked to produce an optical patch, given the same data SAR and optical/SAR from a different day, while the discriminator is trained to judge is such synthesized data is realistic or not. The performance of the scheme is validated using optical data from Landat 8 OLI instrument and Sentinel 2 MSI instrument and SAR data for the Sentinel 1 SAR over regions of diverse land cover types which as used as input to a Random Forest classifier. The experimental results demonstrate that using synthesized imagery in addition to using the available SAR data can lead to almost the same accuracy as the case where both types of observation is available As with [137], in [138] conditional GANs are also considered for missing observations recovery using multitemporal optical and radar (SAR) observations from Sentinels 2 and 1, respectively.

### 5.3. Discussion

An overview of exemplary applications of DNN in the problem of remote sensing observation restoration, which includes the removal of different types of noise and the estimation of missing observations, is presented in Table 6.

A quantified comparison of different denoising methods is presented in Table 7, where gains compared to the state-of-the-art non-DNN BM4D [139] method are presented in terms of PSNR, averaged over the bands. The results indicate that the GAN-based method reported in [121] offers the highest performance gains, while being able to handle different types of noise.

Highest performing methods for restoration involve CNN architectures with multiscale feature extraction capabilities, while GANs architectures also appear promising.While the case of denoising with known noise characteristics has been explored, further research involving unknown or mixed distribution noise is required.The use of multiple sources of observations with different quality characteristics targeting restoration is also another topic of great importance.Estimating missing observation, especially the cases involving cloud occlusion of optical imagery, using radar observations which are impervious to this problem also warrants further investigation.

## 6. Fusion

Fusion refers to the simultaneous encoding of observations from multiple modalities and/or sensors with diverse characteristics, targeting the generation of higher quality or new types of observations [140]. The framework of fusion is closely related to the pan-sharpening one; however, it considers more generic cases such as the joint analysis of high-spectral-low-spatial HS resolution observations with low-spectral-high-spatial MS ones, or observation from optical and radar sensors.

### 6.1. Fusion of MS and HS Observations

Enhancing the spatial resolution of HS observations through fusion with high spatial resolution MS measurements is a major line of research, since HS and MS instruments are characterized by different imaging capabilities due to the inherent trade-offs in the system design [141]. Palsson et al. [142] were among the first to consider a 3D CNN for increasing the spatial resolution HS observation, by exploiting the high spatial resolution of registered MS observations. The method operates on image patches and involves the singular value decomposition to the HS observations so that only the spectral loadings are super-resolved, while preserving the spectral singular values. To train and validate the performance of the CNN, HS observations from the ROSIS Pavia center data set are appropriately downscaled. Experimental comparison with a wavelet-based approach demonstrate that higher quality reconstruction is possible, even when the input observations are affected by noise.

Yang et al. [143] recently proposed a two-branch CNN architecture where features from the HS and the MS are initially extracted independently and then are fused together, effectively enhancing the spectral dimension of MS by transferring the spectral features from the HS. The overall architecture involves sequences of convolutional layers, 1D for the HS and 2D for the MS, and the subsequent concatenation of their outputs which are used as input to a sequence of fully connected layers producing the super-resolved spectrum for each spatial location. The performance is quantified using synthetic observations from the AVIRIS dataset and EnMAP, as well as real observations captured by Sentinel 2 for the MS and the EO-1 Hyperion sensor for HS.

The use of CNN for MS and HS image fusion is further explored in [144] where two sub-networks, an encoder and a pyramid fully convolutional decoder network, are considered for progressively reconstructing the high spatial resolution HS image. The proposed architecture seeks to extract features from multiple scales by introducing different spatial resolution HR imagery during the estimation process, while to further improve the reconstruction quality, the typical $\ell_{2}$ loss is augmented with a term focusing on the gradient difference for generating enhanced details. Experimental validation of the method using simulated observations produced by real Hyperion, HYDICE and ROSIS sensor data demonstrate significant improvements as high as 1 dB for Hyperion, 2.5 dB for HYDICE and 1.5 dB for ROSIS compared to [142] which achieve the second-best performance. Furthermore, the results also demonstrate that the introduction of a more appropriate loss function always have a positive effect in reconstruction quality.

In addition to the fusion of observations with different spatio-spectral characteristics, fusion is also considered for generating high spatio-temporal resolution observations [145]. The proposed architecture is composed of three parts, the spatial resolution enhancement of high-temporal-low-spatial-resolution imagery, the extraction of high-frequency components from low-temporal-high-spatial-resolution observations and the final fusion of the extracted features. Effectively, the architecture is trained by considering pairs of different resolution observations from the same date and using the learned model to enhance the spatial resolution of a low-spatial-resolution observation from a future date. The method is validated using daily observations at 500 m spatial resolution observation from MODIS and 30 m resolution from Landsat 8 OLI sensor which has a 16 day revisit cycle, operating on single bands.

Building on the fusion context, the authors of [146] propose a blind image quality assessment-flavored model for deriving high-resolution shortwave infrared (SWIR) images, via the help of pan-sharpening and hyper-sharpening techniques. More precisely, they build four different architectures which combine PAN and SWIR images to derive very high-resolution images obtained from the WorldView-3 satellite dataset. These approaches (combining widely used pan-sharpening and hyper-sharpening algorithms) differ from each other to the "nature" of the employed fusion process (i.e., sequential, parallel, sequential-parallel, parallel-sequential), and are evaluated with respect to a new image quality assessment measure which weights the level of spectral and spatial distortions (capitalizing on the famous SSIM and Natural Image Quality Evaluator (NIQE) image processing performance metrics). The four above approaches (each one comprising of 10 different algorithmic schemes for the pan-sharpening) are compared in terms of the aforementioned quality measure, as well as of the required computational time, ending up with quite promising results in the context of blind assessment of pan-sharpened images in the case of no-overlapping between the SWIR bands and the PAN one.

### 6.2. Fusion of Spectral and Radar Observations

Extending the majority of fusion approaches which consider MS and HS observations, a more generic fusion framework which explores the fusion of optical sequences, synthetic aperture radar (SAR) sequences and digital elevation model is proposed by Scarpa et al. [147] in an effort to estimate missing optical features, typically due to cloud coverage. The proposed CNN architecture accepts very small patches of cloud-free co-registered optical and SAR images from the Sentinel 2 and 1 platforms, respectively, and is tasked at estimating biophysical parameters such as then normalized difference vegetation index (NDVI). To validate the performance, the quality of the NDVI estimation is quantified over an agricultural region of Burkina Faso, West Africa, and over an extended period of time. Results demonstrate both that high-quality estimation of spectral features is possible from radar data and that CNN-based architectures can handle large temporal gaps between observations.

A key challenge when trying to fusion observations from multiple sources is the registration or equivalently the identification of corresponding patches, between different modalities. To that end, the fusion of optical and radar (SAR) observations is explored in [148] where a so-called “Siamese” CNN network architecture is employed for predicting if patches from the two sources are a match or not. To validate the performance of the system, the authors employ an automated framework called “SARptical” [149] which is able to extract 3D point clouds from optical and SAR observation and perform the matching in this highly accurate 3D point cloud space. Despite the significant challenges due to the difference in viewing angles and the imaging protocol employed by SAR systems, the results that the quality of matching is comparable to state-of-the-art hand-crafted features.

In [150], Quan et al. consider the use of GANs for generating examples for registration of optical and SAR imaging data. Since both optical and SAR observations of a scene may not even be available, traditional data augmentation techniques for deep learning-based models -employing geometric transformations for generating abundant training data- are out of the question for the problem at hand. To overcome such an obstacle, the authors propose the use of a GAN which is charged with the task of producing coupled training data with respect to its input (i.e., derive pseudo SAR data when fed with optical data, and vice-versa). Subsequently, a CNN is employed to infer the labels of the generated multi-modal image data (in a patch-wise level). At the test phase, the trained model in conjunction with three different kind of constraints that need to be met, is used for predicting the labels of the test images and the transform matrix alongside with the registered images as well. The proposed method is tested with several registration methods in different multi-modal image data (i.e., optical-and-SAR, optical-and-LIDAR), outperforming them in several qualitative and machine learning-based measures.

The generation of SAR observations which although highly realistic do not match corresponding optical examples is explored in [151], in an effort to generate challenging negative examples which can lead to higher matching accuracy and lower false positive rates between SAR and optical observations presented in [148]. The Conditional GAN framework is also explored in [152] for the improvement of the geolocalization accuracy of optical imagery through the automatic extraction of ground control points from SAR derived imagery. The approach involved using cGANs for generating SAR-like images from optical imagery, which can then be compared to actual SAR observation and used to extract ground control points which are used by matching. Once the matching parameters are found, they can be directly applied to the original optical imagery to achieve high-quality registration. The proposed method is validated through the demonstration of increased matching accuracy on observations from TerraSAR-X and PRISM measurements.

### 6.3. Discussion

Table 8 provides a quick outlook at the different DNN-based approaches in remote sensing observation fusion. We note that we focus on the case where observations from multiple sources are fused to provide high quality and/or more informative observations. Given the existing state-of-the-art, the following observation can be made.

In Table 9, the performance of three MS/HS fusion methods, two specifically developed for fusion [142,144], and a generic pan-sharpening one [76] with respect to PSNR are presented. These results indicate the problem of HS/MS fusion is different compared to pan-sharpening and that more case-specific methods, for example [144] which considers the fusion of features from both modalities, offer better performance.

Almost exclusively, CNN architectures have been used for HS and MS fusion, the majority of which follow the same principles as the case of pan-sharpening.Although different approaches have been proposed for addressing pairs of resolution dimensions, i.e., spatial-spectral and spatial-temporal, no approach has been put forward for increasing the resolution along spatial, spectral and temporal resolution.There is limited investigation in fusion of observation from different modalities, i.e., optical and radar. We believe this domain to be extremely promising and hence more research needs to be conducted.

## 7. Challenges and Perspectives

The impact of machine learning in remote sensing observation analysis has been of paramount importance, since the first application of such methods in the late 1990s, primarily focusing on the automated extraction of information such as land cover estimation [153] and detection of oil spills [154]. Since the early 2000, machine learning has been gaining attention for problems related to observation enhancement such as super-resolution [155], while in the past five years, this problem has been radically addressed through DNN approaches, demonstrating that machine learning and DNN is particular have still significant unexplored potential in the remote sensing domain.

Karpatne et al. [156] present a comprehensive investigation of the potential and challenges of machine learning in geosciences, a key application domain of remote sensing, which involve: (i) inherent challenges associated with geoscience processes such as spatio-temporal characteristics and multi-variate nature of phenomena, (ii) challenges associated with geoscience data collection including different sampling resolutions, missing measurements and highly varying quality and (iii) scarcity of training examples and associated ground truth.

One of the major benefits of DNN is their ability in *simultaneously addressing multiple problems* [157]. For example, in [158], the authors propose a multiscale fully convolutional network for multi-task problems and more specifically for the simultaneous super-resolution and colorization of remote sensing imagery. Another potential area of innovation involves the exploration of enhancing satellite derived products such as land surface temperature and soil moisture. As an illustrative example, while typical remote sensing super-resolution approaches are applied to imagery, may that be PAN, MS or HS, CNNs [34] have also been recently considered for super-resolving climate variables such as daily precipitation measurements [159] surpassing in performance more established approaches.

From an architectural point-of-view, undoubtedly, one of the most interesting and innovative approaches is the concept of GANs. In the previous sections, the application of GANs in different problems are presented, including super-resolution [64], pan-sharpening [116] and restoration [127]. In addition to the potential of GANs in well-known problems, GANs have also been proposed for addressing emerging problems. A particular instance involves the generating realistic training examples. as in the case of [160], where Lin et al. explore the potential of generating realistic remote sensing imagery by training a GAN to map images for segmentation purposes, while in [161], GANs are employed for generating ground-level views from aerial and satellite imagery.

While using machine learning for the automated extraction of actionable information from a single source has been extensively studied, substantially less effort has been allocated on approaches for jointly analyzing observations from *multiple modalities* and *different sources*. Compared to the singe-modality case, handling observations from multiple instruments can lead to a much higher quality estimation of geophysical parameters, exploiting each instrument’s unique capabilities. Achieving this objective requires the simultaneous encoding of diverse types of observations where each spatial location on the Earth for example, must be associated with a spectral profile encoding the response over multiple wavelengths acquired with MS, and reflected signals of different polarizations acquired by SAR. Enabling this capability requires the ability to simultaneously encode high-dimensional observation, more than 3 which is currently the state-of-the-art.

Another, equally important challenge characterizing existing approaches, is the inability of integrating observations associated with *diverse sampling scales*. In the case of remote sensing, this translates to integrating low spatial resolution global-scale satellite data with high-accuracy localized in situ sensor network measurements. A consequence of paramount importance related to this issue is that the systematically acquired satellite observations require human experts for providing annotations, which, although of high quality, cannot be effectively scaled-up. To address this challenge, coarse satellite observations will need to be automatically annotated using “ground-truth” measurements from in situ networks and ground-based surveys, seamlessly handling the variety in spatial and temporal scales, and the irregularity of sensing patterns. Some initial attempts have been explored for remote sensing observation classification e.g., [7]; however, more research is needed in order to enable the automated accurate and timely estimation of key environmental parameters at global scales.

Analyzing observations from a single time instance cannot provide the necessary insight into the temporal dynamics of phenomena of interest. *Time series analysis* can provide concrete solutions to problems such as estimating measurements from regions where no up-to-date observations are available, effectively increasing the temporal resolution. Furthermore, data-driven time series processing can enable the prompt identification of anomalies, a crucial issue since subtle changes within normal variation, which cannot be identified using pre-defined thresholds, can indicate the onset of major environmental events. Multitemporal remote sensing observation classification is gaining traction, e.g., [162]; however, addressing the challenges associated with observation enhancement is still in its infancy and can only be achieved by automatically exploiting correlations across modalities and time.

A last yet quite important point is the incorporation of prior knowledge in DNN models, a very active research topics which seeks ways of introducing physics guided modeling into the design of neural networks. Indeed, the use of highly accurate and physical plausibility constraints are key ingredients when trying to use DNN for scientific discovery, especially for the case of Earth monitoring [163]. Currently, the few research papers published on the topics consider the case of dynamical systems where in addition to a typical loss function, an additional regularization penalizing physical inconsistency is introduced. For example, in the work by Karpatne et al. [164,165], the output of a physics-driven model of lake temperature is considered to be extra features, in addition to actual drivers such as meteorological conditions, in the DNN algorithm, while a physics-based loss function is also considered during the optimization process. Although this framework has not been considered for the case of remote sensing observation analysis and enhancement, we expect that significant benefits can results from the integration of data-driven and physics-driven models, where for example, one could consider the physical processes governing soil dynamics during a dynamic observation super-resolution process.

## Figures and Tables

**Figure 1 sensors-19-03929-f001:**
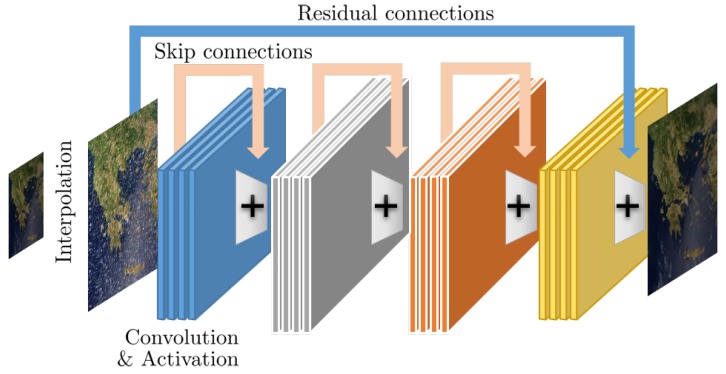
A typical CNN architecture for remote sensing image enhancement featuring convolutional and nonlinear activation layers with residual and skip connections.

**Figure 2 sensors-19-03929-f002:**
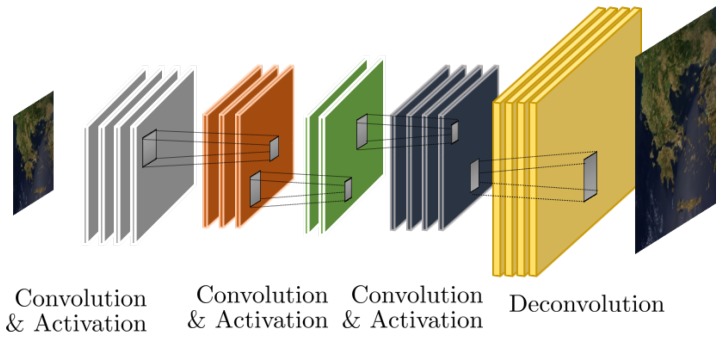
*Hourglass* shaped CNN architecture.

**Figure 3 sensors-19-03929-f003:**
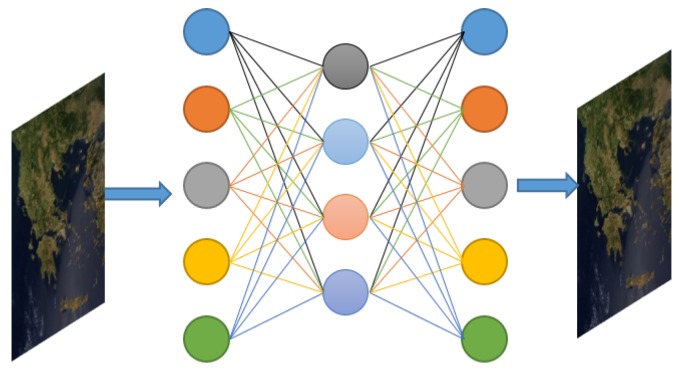
Typical structure of a single-layer autoencoder: This one-hidden layer structure learns the best possible compressed representation, so that the output is as close as possible to the input.

**Figure 4 sensors-19-03929-f004:**
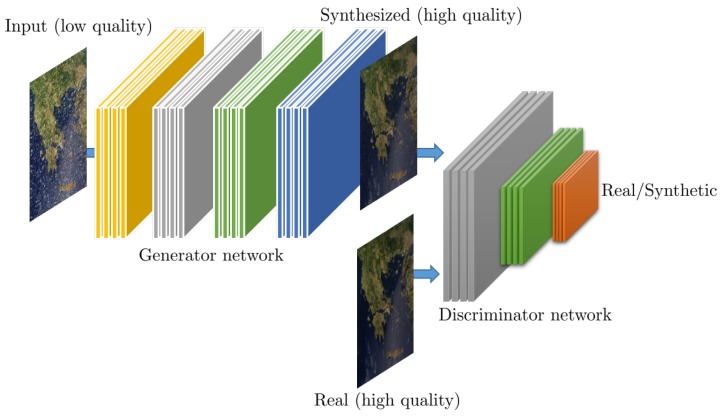
A (conditional) Generative Adversarial Network architecture composed of a Generator and a discriminator network for image quality enhancement.

**Figure 5 sensors-19-03929-f005:**
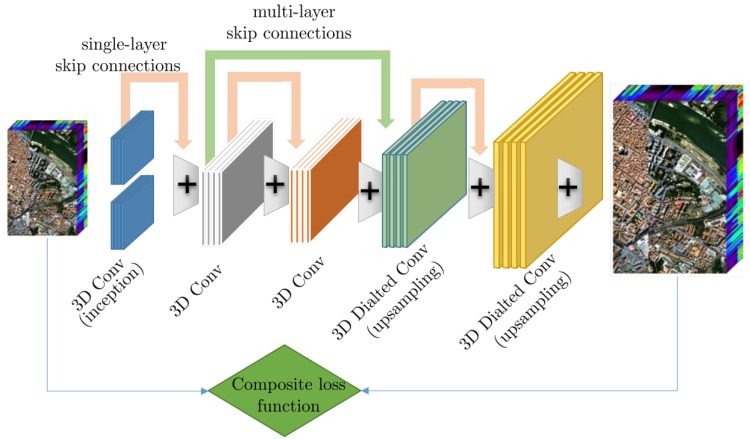
An exemplary CNN architecture for MS/HS super-resolution.

**Figure 6 sensors-19-03929-f006:**
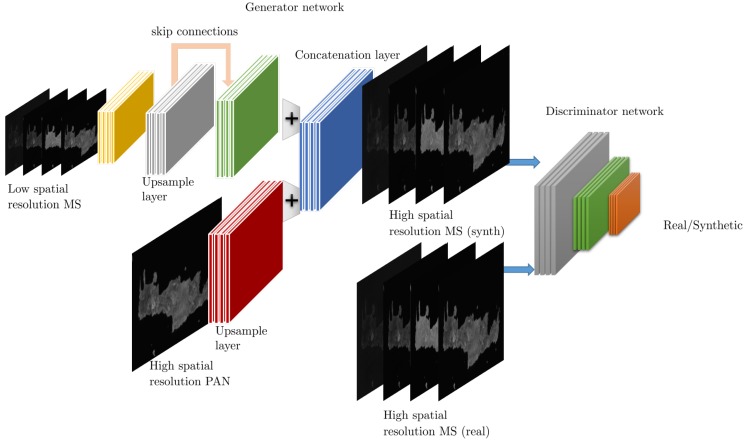
An exemplary two-stream GAN-CNN architecture for pan-sharpening.

**Table 1 sensors-19-03929-t001:** Listing of representative super-resolution approaches.

Methods	Observation Type	Approach
[51,52,53]	Single-Image/band	single-scale CNN
[54,55,56,57,59]	Single-Image/band	multiscale CNN
[62,63,64]	Single Image	GAN
[68]	HS/MS	multiscale CNN (LPN)
[72]	HS/MS	multiple loss CNN
[77]	HS/MS	band-specific CNN
[78]	HS/MS	3D-CNN
[91,92,93]	Video frames	CNN variants

**Table 2 sensors-19-03929-t002:** Relative performance gains (with respect to PSNR) for different DNN-based *single image* super-resolution methods compared to bicubic interpolation for scale factors ×2, ×3 and ×24.

Method	Dataset	Performance Gain		
		×2	×3	×4
LGCNet [54]	UC Merced	+9%	+7%	+5%
WMCNN [56]	RSSCN7	+8%		+3%
[62]	UCMERCED/RSSCN7/NWPU-RESIS45	+9%		+7%
TGAN [63]	UC Merced (airplanes) pre-trained on the DIV2K			+16%
DMCN [55]	UC Merced	+10%	+8%	+7%
MRNN [59]	NWPU-RESISC45			+13%
FRDBPN [60]	UC Merced			+5%
EEGAN [64]	Kaggle	+14%	4%	+13%

**Table 3 sensors-19-03929-t003:** Relative performance gains (with respect to mean PSNR) for different DNN-based *multispectral* super-resolution methods compared to bicubic interpolation for scale factors ×2, ×3 and ×24.

Method	Dataset	Performance Gain		
		×2	×3	×4
[68]	UC Merced			+2%
MLFN [72]	UC Merced	+11%		
SSIN [77]	UC Merced	+8%	+6%	+4%
3D-FCNN [78]	UC Merced	+9%	+5%	+3%

**Table 4 sensors-19-03929-t004:** Listing of representative pan-sharpening approaches.

Method	Approach
[97,98,100,101]	AE variants
[76,102]	CNN based on SRCNN architecture
[99,103,104,105,106]	CNN with residual connections
[107,108]	Inception-like CNN for multiscale feature extraction
[110]	Target-adapted CNN
[111,112]	Two-stream CNN architecture
[113]	CNN-based pixel-level regression over multiple scales
[114]	Two CNNs associated with spectral and spectral dimension
[116]	Two-stream GAN-CNN architecture

**Table 5 sensors-19-03929-t005:** Relative performance gains (with respect to SAM) for different DNN pan-sharpening methods compared to non-DNN MTF-GLP [118] method and baseline DNN-based PNN [76] method.

Method	Dataset	Baseline Approaches	
		MTF-GLP	PNN
PUNET [110,113]	Ikonos/WorldView-2	-	+15%/+16%
DRPNN [99]	QuickBird/WorldView-2	+35%/+25%	+8%/+5%
MSDCNN [107]	QuickBird/WorldView-2	+35%/+26%	+7%/+5%
RSIFNN [111]	Gaofen-1/QuickBird	-	+11%/+8%
L1-RL-FT [110]	WorldView-2/WorldView-3	+38%/+29%	+14%/+37%
BDPN [112]	QuickBird/WorldView-3	-	+25%/+ 11%
[115]	WorldView-2	-	+33%
[114]	Pleiades/WorldView-2	+50%/+26%	+46%/+20%
PNN [76]	WorldView-2/Ikonos	+25%/+21%	-
PanNet [106]	WorldView-3	-	+15%
PSGAN [116]	QuickBird/GF-1	-	+18%/+35%

**Table 6 sensors-19-03929-t006:** Major approaches in DNN-based remote sensing observation restoration.

Method	Observation Type	Approach
[120]	HSI + Gaussian noise	CNN with trainable activation function
[121,122]	HSI + Gaussian noise	Residual CNN with key band selection
[123]	Aerial RGB + Gaussian noise	Multiscale CNN for learning residual noise component
[124]	HSI + Gaussian noise	Multiscale 3D CNN using neighboring bands
[125]	HSI + Gaussian/Poisson noise	Multiscale 3D CNN w/atrous layers
[127]	HSI + mixed noise	2D CNN with residual connection
[127,128]	Noise & Blur	GAN architecture
[131]	Missing spatial measurements	CNN-based fusion using auxiliary observations
[132,133]	Missing temporal measurements	CNN for recovery via temporal prediction
[134]	Cloud occlusions	GAN
[137]	Missing MS observations	cGAN using multitemporal MS and SAR observation

**Table 7 sensors-19-03929-t007:** Relative performance gains (with respect to mean PSNR) for different DNN denoising methods compared to non-DNN BM4D [139] method.

Method	Dataset	Comparison to BM4D [139]
HDnTSDL [122]	Washington DC Mall (HYDICE)	+12%/+10%/+13% (Gaussian w/ σ = 15/25/50)
SSDRN [121]	PAVIA (ROSIS)	+2%/+3%/+4% (Gaussian w/ σ = 15/25/50)
SSGN [126]	Pavia (ROSIS), Washington DC Mall (HYDICE), Indian Pines (AVIRIS)	+20%/+17% (Gaussian, Gaussian+stripe)
[124]	Washington DC Mall (HYDICE)	+6%/+8%/+11% (Gaussian w/ σ = 25/50/100)
[125]	Pavia University (ROSIS), Indian Pines (AVIRIS)	+1%/+4% (Gaussian w/ σ = 10)

**Table 8 sensors-19-03929-t008:** Listing on key approaches in DNN-based observation fusion.

Method	Inputs	Objective	Approach
[142]	MS and HS	spatial/spectral resolution	3D-CNN using concatenated observations
[143,144]	MS and HS	spatial/spectral resolution	Fusion using two-stream CNNs
[145]	MS and HS	spatial/temporal resolution	CNN sub-networks fusion
[147]	MS and SAR	NDVI estimation	CNN using concatenated observations
[148,150]	RGB and SAR	registration	GAN-based framework

**Table 9 sensors-19-03929-t009:** Relative performance gains (with respect to PSNR) for three DNN-based MS/HS fusion approaches.

Method	Dataset			
	Botswana (NASA EO-1)	Washington DC Mall (HYDICE)	Pavia University (AVIRIS)	
PFCN + GDL [144]	35.36	38.38	41.80	
3D-CNN [142]	34.03	36.48	39.93	
PNN [76]	30.30	28.71	36.51	

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
