# Peer review of "Survey of Deep-Learning Approaches for Remote Sensing Observation Enhancement"

_sensors, 2019, doi:10.3390/s19183929_

Round 1
Reviewer 1 Report
The paper is interesting and well-organized and well-written.
Minor comments
It would be useful to extend Table 1, Table 2, Table 3, Table 4 with a column summarysing the performance of the described approach with respect to state-of-the-art methods.
Minor edits
pag. 6 line 209
for tied weights, V should be equal to transpose of W but is V=W
pag. 9 line 297
"most bands for bands in the visible" should be "most bands in the visible"
pag. 10
section name "2.1.1. Single image image super-resolution" image is duplicated
pag. 27 line 1091
I think "be that may" should be "may be that"
Author Response
We would like to thank the reviewer for the valuable comments. All text based suggestion have been addressed, while five additional tables (current pdf version Tables 2,3,5,7 and 9) have been included which provide numerical performance gains associated with representative approaches.
Reviewer 2 Report
I am a picky reviewer, and this is the first time in many, many years that I recommend a new manuscript to simply be published AS IS. Kudos to the authors for writing an outstanding review paper that 1) fills an important gap in literature; 2) is extremely comprehensive in all the topics it covers; and 3) is well organized and well written, with great graphics and ample references for readers to dig deeper.
In particular, I liked that this paper does not only review and point to the literature on the various topics, but provides excellent, intuitive explanations of major concepts right there, as well as practical guidelines of how to address specific challenges. It presents thus a very useful, directly applicable guide for the community (including myself). Furthermore, the topic is well chosen - while there are countless papers on how to use neural networks for classification and prediction tasks in remote sensing, there is indeed very little literature on the topic of remote sensing enhancement, and this paper fills this important gap.
Honestly, I am just hoping that this paper will be published very quickly, so that I can start using it immediately in my own work, as well as freely share it with my colleagues who are also working on remote sensing tasks. Thank you for such a great contribution to the community. I believe this review can play a major role in disseminating what is known about the topic and moving the field forward.
I have a few thoughts for the authors, but those are just suggestions, maybe for future work.
1) Receptive fields: Do you have any thoughts on how to use the analysis of receptive fields more broadly for the design of ANNs for image enhancement? You mention receptive fields in Section 4.1, but I think they might have broader use for network design.
2) Do you think the field of physics-guided machine learning (PGML) might be useful for fusion tasks?
Thanks again for such a great contribution. The community needs more papers like this one.
Author Response
We would like to express our gratitude for the warm comments made by the reviewer. We too hope this work will prove to be a valuable asset for researchers in this fast change landscape. We would also like to especially thank the reviewer for pointing us into the topic of PGML which indeed appears to be very promising. We were unable to locate any paper on the topic of fusion, but we completely agree that there is great potential, so we have included some discussion on the topic.
Regarding the two points raised, we have introduce the follow two texts in the manuscript in Sections 1.1.1. and Section 6 respectively.
Receptive fields
A key characteristic associated with CNN is the receptive field, a termed borrowed from the biology which refers to the part of the sensed space that can elicit some neural response. Receptive fields apply primarily to the case of CNN and refer to the spatial extend associated with each convolution kernel, while in the majority of CNN architectures for image enhancement, both generic and remote sensing ones, the convolution kernels are of size 3x3, and in some cases 5x5 or even 7x7. However, due to the hierarchical structure of CNNs, each subsequent layer has a larger effective receptive field compared to the previous layer, allowing the CNN to progressively consider larger and larger parts of the input image. This process increases the receptive field by a factor of two for each additional layer. Dilated convolutions can also be utilized for increasing the receptive field by skipping pixels during their applications, e.g., a 3x3 kernel is effectively applied on a 5x5 region by padding, allowing for faster rates of increase. Not that pooling operator also increase the receptive field, however, they are not typically used in inverse problem due to the unwanted reduction is the output size. In terms of image enhancement, it has been shown that increasing the size of the receptive field, e.g., through the use of dilated convolutions, can lead to the extraction of more important features for generic image132denoising [17] and super-resolution [18], as well as in despeckling SAR observations [19].
physics-guided machine learning
A last yet quite important point, is the incorporation of prior knowledge in DNN models, a very active research topics which seeks ways of introducing physics guided modeling into the design of neural networks. Indeed, the use of highly accurate and physical plausibility constraints are key ingredients when trying to utilize DNN for scientific discovery, especially for the case of Earth monitoring [161]. Currently, the few research papers published on the topics consider the case of dynamical systems where in addition to a typical loss function, an additional regularization penalizing physical inconsistency is introduced. For example, in the work by Karpatne et al. [162,163], the output of a physics-driven model of lake temperature is considered as extra features, in addition to actual drivers like meteorological conditions, in the DNN algorithm, while a physics-based loss function is also considered during the optimization process. Although this framework has not been considered for the case of remote sensing observation analysis and enhancement, we expect that significant benefits can results from the integration of data-driven and physics-driven models, where for example, one could consider the physical processes governing soil dynamics during a dynamic observation super-resolution process.